Dominance in dogs as rated by owners corresponds to ethologically valid markers of dominance

Kubinyi Enikő eniko.kubinyi@ttk.elte.hu
Wallis Lisa J.
Department of Ethology, Eötvös Loránd University , Budapest , Hungary
Hopper Lydia
Electronic publication date: 2019 May 9
Publication date: 2019
Volume: 7
Electronic Location ID: e6838
Received 2018 Nov 21; Accepted 2019 Mar 24
Copyright: ©2019 Kubinyi and Wallis
Copyright year: 2019
Copyright holder: Kubinyi and Wallis
License: This is an open access article distributed under the terms of the Creative Commons Attribution License, which permits unrestricted use, distribution, reproduction and adaptation in any medium and for any purpose provided that it is properly attributed. For attribution, the original author(s), title, publication source (PeerJ) and either DOI or URL of the article must be cited.
License URL: https://creativecommons.org/licenses/by/4.0/

Keywords: Dominance, Domestic dog, Animal personality, Submission, Agonistic behavior, Ageing, Leadership

Funding: European Research Council (ERC) under the European Union’s Horizon 2020 research and innovation program Grant Agreement No. 680040 Bolyai+ ÚNKP-18-4 New National Excellence Program of the Ministry of Human Capacities This project has received funding from the European Research Council (ERC) under the European Union’s Horizon 2020 research and innovation program (Grant Agreement No. 680040), from the János Bolyai Research Scholarship of the Hungarian Academy of Sciences, and from the Bolyai+ ÚNKP-18-4 New National Excellence Program of the Ministry of Human Capacities. The funders had no role in study design, data collection and analysis, decision to publish, or preparation of the manuscript.

==============================
Dominance is well defined in ethology, debated in psychology, and is often unclear among the dog owning public and in the press. However, to date, no study has examined how owners perceive dominance in dogs, and what different behaviours and personality types are used to describe dominant and subordinate individuals. A questionnaire study was launched to investigate the external validity of owner-derived estimates of dominance in dog dyads sharing the same household (N = 1,151). According to the owners, dogs rated as dominant (87%) have priority access to resources (resting place, food, and rewards), undertake certain tasks (defend and lead the group, bark more), display dominance (win fights, lick the other’s mouth less, and mark over the other’s urine), share certain personality traits (smarter, more aggressive and impulsive), and are older than their partner dog (all p < 0.0001). An age-related hypothesis has been suggested to explain dominance in dogs; but we found that dog age did not explain the occurrence of dominance related behaviours over the owners’ estimate of dominance status. Results suggest that owner-derived reports of dominance ranks of dogs living in multi-dog households correspond to ethologically valid behavioural markers of dominance. Size and physical condition were unrelated to the perceived dominance. Surprisingly, in mixed-sex dyads, females were more frequently rated as dominant than males, which might correspond to a higher proportion of neutered females in this subgroup. For future studies that wish to allocate dominance status using owner report, we offer a novel survey.

Introduction

The term dominant is often used by dog owners to describe dogs; however, there may be little agreement regarding its meaning, as dominance is defined differently in ethology, psychology, and among the public. In ethology, dominance describes long-term dominant-subordinate social relationships within a dyad or group (Clutton-Brock et al., 1979; Drews, 1993). Dominant individuals usually have priority access to key resources such as food and reproductive partners, due to the consistent winning of agonistic interactions or deference, during which one individual consistently gives way to another (Lorenz, 1966; Smith & Price, 1973). However, in psychology, dominance is often referred to as a personality trait (Gosling & John, 1999) and describes the disposition of an individual to assert control in dealing with others. Finally, the word “dominance” is defined as having control, authority, and power or influence over others (Westgarth, 2016), and the general public may use this word to describe individuals who are more powerful, successful, or important than others. When we consider these three definitions, it is not surprising it is unclear what dog owners mean when they use the term ‘dominance’ in reference to their dogs.

In the next paragraphs we summarize the current knowledge about dominance in dogs and then we examine how scientific findings are related to the perception of dominance in the dog owning public. Although dominance hierarchies have previously been described in free-ranging dogs (Bonanni et al., 2010; Cafazzo et al., 2010; Bonanni & Cafazzo, 2014), in dogs living in packs in enclosures (Range, Ritter & Viranyi, 2015; Van Der Borg et al., 2015; Dale et al., 2017), and in neutered pet dogs at a dog day care centre (Trisko & Smuts, 2015; Trisko, Smuts & Sandel, 2016), the existence and validity of linear dominance hierarchies in these animals is highly debated both by the public and some researchers, mainly because agonistic interactions are rare and contextual (Schilder, Vinke & Van der Borg, 2014). Data on kennelled dogs suggest that dominance is based on submission (signalled mostly by body tail wag and low posture) rather than on aggression (Van Der Borg et al., 2015). Therefore, it has been suggested that domestication has altered the social behaviour of dogs compared to wolves, and submissive behaviour is used to defuse conflicts (Bradshaw, Blackwell & Casey, 2009).

In addition, as Van Kerkhove (2004) notes, although dominance hierarchies in dogs are often described through access to resources (or “competitive ability”, (De Waal, 1986)), not all individuals are equally motivated (or physically able) to obtain them. Therefore the subjective resource value, in combination with associative learning (Bradshaw, Blackwell & Casey, 2009; Bradshaw, Blackwell & Casey, 2016) and personality (McGreevy et al., 2012) can explain interactions between dogs more simply than dominance theory. Moreover, if researchers do not assume the existence of a dominance hierarchy, they seldom identify one, thus a more dynamic approach is needed in order to understand social organizations (Overall, 2016).

However, when a hierarchy was detected in a dog group, several parameters have been shown to covary with dominance status, such as age, sex, and personality. Older dogs were found to be more often dominant than young individuals (Mech, 1999; Peterson et al., 2002; Bonanni et al., 2010; Bonanni et al., 2017; Cafazzo et al., 2010; Trisko & Smuts, 2015). Therefore Bradshaw, Blackwell & Casey (2016) suggested that a simple rule of thumb could help to explain formal dominance in dogs: “in order to be allowed to stay in the group, perform affiliative behaviour towards all the members of the group older than you are”. However, in a group of domestic dogs, van der Borg and colleagues (2015) did not find correlations of rank with age, and it remains unexplored whether the age related hypothesis is a better predictor of formal dominance than dominance displays.

Concerning sex as a potential confounding factor of dominance, conflicts between dogs living in the same household are more common between dogs of the same sex, and female–female pairs are most often affected (Sherman et al., 1996; Wrubel et al., 2011). Mixed-sex dyads are more likely to affiliate and less likely to show unidirectional displays of submissiveness and aggression than same-sex pairs (Trisko & Smuts, 2015). In wolves, separate male and female age-graded dominance hierarchies have been observed in captive packs (Packard, 2003). Overall, male wolves were found to be more often dominant and/or leaders of the pack (Clark, 1971; Haber, 1977; Mech, 1999). In one study on free-ranging dogs, a sex age-graded hierarchy was found, such that males dominate females in each age class, and adults dominate over subadults, and subadults over juveniles (Cafazzo et al., 2010). However, sex had no clear effect on dominance in a family pack of captive arctic wolves, although sex-separated linear hierarchies showed a stronger linearity than female-male hierarchies (Cafazzo, Lazzaroni & Marshall-Pescini, 2016).

Personality traits might also associate with dominance status. For example, aggression towards people and controllability was linked to dominance rank and leadership in pet dogs according to Ákos et al. (2014). Since some dog owners describe dogs that often show dominant behaviour towards other dogs as having a “dominant personality”, studies linking personality traits to dominance status would be especially useful to help clarify the correct terminology to the public. Dog owners confusion regarding the term dominance can be partly explained by the fact that based on a literature review on canine personality, researchers have identified a broad dimension labelled as ‘Submissiveness’, and defined it as the opposite of dominance (Fratkin et al., 2013). According to the authors, “Dominance can be judged by observing which dogs bully others, and which guard food areas and feed first. Submission can also be reflected by such behaviours as urination upon greeting people”. Thus, even in the scientific literature some authors define dominance as a personality trait, and there is an ongoing debate in human, primate, and dog personality research on how to interpret certain traits. However, according to the majority of ethologists dominance is not a personality trait (Schilder, Vinke & Van der Borg, 2014). While personality is largely independent of context and is stable over time (Jones & Gosling, 2005) dominance status depends on the interacting partners.

The popular media has also played a role in influencing owners’ attitudes, by often describing dominant dogs as those with behavioural problems or a tendency towards aggression. A dog is often referred to as dominant when it “misbehaves”, e.g., jumps up on or shows aggression towards the owner. The belief that such behaviours may signify that the dog is attempting to control the owners’ behaviour, is based on erroneous models of wolf pack organisation, and has often been used to justify the use of abusive training techniques (Bradshaw, Blackwell & Casey, 2016). However, negative reinforcement and positive punishment training techniques can cause increased stress, fear and mistrust, and are associated with increased aggression towards other dogs in the household (Casey et al., 2013), and towards human family members (Casey et al., 2014). Positive punishment has obvious abusive connotations too, according to these studies.

Previously, several studies have attempted to determine the dominance rank of dogs living in multi-dog households by utilising owner questionnaires (Pongrácz et al., 2008; Pongrácz, Bánhegyi & Miklósi, 2012; Ákos et al., 2014). Pongrácz et al. (2008) used a four item questionnaire to measure dogs’ dominance status in dyads, and related them to differences in social learning in response to a human or dog demonstrator. Dogs were identified as dominant if they displayed at least three behaviours out of four (barked more/longer, licked the other dog’s mouth less, ate first, and won fights). Dominant dogs were less likely to learn from observing other dogs, and more likely to copy a human demonstrator. They also performed better than subordinates in a problem solving task, but only when observing a human demonstrator (Pongrácz, Bánhegyi & Miklósi, 2012). Subordinate dogs showed better learning in the dog demonstrator condition. Results from both studies suggest that social rank affects performance in social learning situations, as dominant dogs tend to follow humans while subordinate dogs follow other dogs. Thus, owner questionnaires could be a valid method to determine the dominance rank of individuals within dog dyads, similarly to other dog behaviour studies, particularly as the quality of data produced by citizen scientists has proved to be satisfactory (Hecht & Spicer Rice, 2015).

To understand how the dog-owning public use the word “dominance”, we evaluated what attributes they associated with dominance using a questionnaire study. We have to note here, that, as Westgarth suggests (Westgarth, 2016), it is possible that a dominance hierarchy is not fundamental to the structure of the dogs’ social system, but is rather the by-product of human observation. According to this view, dominance is simply the question of individual interpretation, and this is another reason to investigate how the public interprets “dominance” in dogs. In this study, we surveyed people that owned multiple dogs. We investigated the relationship between the dogs’ ranks, behaviour, and demography. We were interested in finding out whether dogs that the owners have classified as “dominant” display certain behaviours more or less often than their subordinate partner. We also tested the age related hypothesis suggested by Bradshaw et al. (2016) by determining which factor best explained behavioural and demographic differences between the dyads, owner reported hierarchical status or age status.

Materials and Methods

Ethical statement

The procedures applied complied with national and EU legislation and institutional guidelines (ethical approval: Government Office for Pest County PE/EA/2019-5/2017). Participants were informed about the identity of the researchers, the aim, procedure, and expected time commitment of filling out the survey. Owners filled out the survey anonymously; therefore, we did not collect personal data. Participants could at any point decline to participate (Supplemental Information S1).

Subjects

Between 25th June and 13th August 2017, 1156 owners of at least two dogs filled in a questionnaire in Hungarian, which was advertised in a social media Dog Ethology group. We identified the dogs using their given names, to ensure that no duplicate entries were included in the analysis. After data cleaning and deleting of duplicate entries, 1151 responses remained, which detailed owners’ responses for unique individual pairs of dogs. Owners indicated the sex and reproductive status of each dog in the dyad, after allocating them to either Dog A or Dog B (based on their own choice). We have no information on how owners chose which dogs to compare if they had more than two dogs. Twenty three percent of the dyads consisted of males only, 28% females only, and 49% were both sexes. The percentages of neutered individuals were 45% in males and 62% in females.

Procedure

The questionnaire consisted of 21 items (Table 1). In the case of items 1–19, owners indicated which of the two dogs best fitted the description: Dog A, or Dog B. Owners could also select “Similar” if both dogs fitted the description, or “N/A”. When the owners marked “N/A” we assumed that they could not answer the question as the dog/dogs did not display that behaviour, or that situation did not occur (e.g., the dogs never fight with each other or they do not go for walks together), or owners were unsure/did not fully understand the question, or the answer was not known to them (e.g., they could not assess which of the dogs was in better physical condition). We chose the behaviours based on previous studies (Pongrácz et al., 2008; Pongrácz, Bánhegyi & Miklósi, 2012), and included markers of agonistic (i.e., winner of fights) and formal dominance (i.e., licking the mouth of the partner, usually during greeting ceremonies (signalling the acceptance of lower social status) (Bonanni et al., 2010), as well as resource-holding potential (obtains more food, (Vervaecke, De Vries & Van Elsacker, 2000), better resting places, etc.). In addition, we included other factors, which have previously been proposed to be relevant when measuring leadership and dominance, such as age, sex, size, physical condition, leadership and specific behavioural characteristics, including intelligence, obedience, aggressiveness and impulsiveness (Drews, 1993; Conradt & Roper, 2003; Conradt & Roper, 2005; Cafazzo et al., 2010). Items 2–4 and 6 were the same as those used in Pongrácz et al. (2008). In the case of items 20 and 21, the owner could also indicate “both” or “neither” dogs (Table 1).

Table 1 Questionnaire items.

Owners were asked to fill out the questionnaire for two of their dogs (‘A’ and ‘B’) and indicate which dog corresponds better to the description. They could also select “Similar” if both dogs fitted the description or “N/A” if the question did not apply to the dog dyad.

Item number	Item name	Questions	
1	status	Which of the dogs is the “boss” (has a dominant status) to the best of your knowledge?	
2a	bark	When a stranger comes to the house, which dog starts to bark first (or if they start to bark together, which dog barks more or longer)?	
3a	lick mouth	Which dog licks the other dog’s mouth more often?	
4a	eat first	If the dogs get food at the same time and at the same spot, which dog starts to eat first or eats the other dog’s food?	
5	reward	If they got a special reward (e.g., a marrowbone), which dog obtains it?	
6a	fight	If the dogs start to fight, which dog wins more frequently?	
7	play ball	If you play with a ball with both dogs, which one retrieves it more frequently?	
8	greet owner	When you enter your home, which dog greets you first?	
9	walk first	Which dog goes in the front during walks?	
10	resting place	Which dog acquires the better resting place?	
11	overmark	Which dog marks over the other’s urination?	
12	defend group	If the dog’s group is perceived as being under attack, which dog is in the front?	
13	smart	Which dog is smarter?	
14	obedient	Which dog is more obedient?	
15	aggressive	Which dog is more aggressive?	
16	impulsive	Which dog is more impulsive?	
17	size	Which dog is heavier?	
18	physical condition	Which dog is in a better physical condition?	
19	age	Which dog is older?	
20	sex	Which dog is male?	
21	neutered	Which dog is neutered?	
Notes.

a Adopted from Pongrácz et al. (2008).

Statistical analysis

Analyses were performed in SPSS 22.0 and R 3.3.2. Descriptive statistics were calculated for the sample and summarised in the results section. Note that we did not have the opportunity to use dominance rating (dominant vs subordinate) as a response variable in a model directly, due to the design of the questionnaire, which collected information for dyads (resulting in one line of data per dyad), and not individual dogs. Therefore individual binomial analyses were the best way to answer our question, “Do dogs which the owners classify as “dominant” show certain behaviours more or less often than their subordinate partner?” and to deal with missing values.

Binomial tests using Dominance Status on the full sample

To investigate the owners’ responses for each item (1 to 21), we calculated the percentage allocation of the dogs to each possible category: “Differ” (the dogs in a particular dyad differed in that behaviour/characteristic), “Similar” (the dogs’ behaviour was similar) and “N/A” (the owner was not able to determine if the dogs differed).

In order to answer the question “do dogs that the owners classify as “dominant” show certain behaviours more or less often than their subordinate partner”, we used binomial tests to compare the distribution of observations between the dogs for each of the replies to items 2 to 21. We included only the dogs that were allocated a “dominant” or a “subordinate” status, based on the response of the owner to item 1 (“Which of your dogs is the boss/dominant). We did not consider dyads where owners indicated that their dogs were “Similar” in dominance status, or where they marked “N/A” (N = 148). (Sample sizes are indicated in Fig. 1 for each item). We lowered the p level to 0.0023 from 0.05 as suggested by a Bonferroni correction for the 22 comparisons.

Figure 1 Differences between characteristics in dog dyads.

Colours: orange: dogs in a dyad differ in the characteristic, yellow: dogs are similar, green: N/A. Sample sizes are indicated on the columns, item numbers are in brackets. Seven items, highlighted by *, are suggested for future work based on both their significant link with dominance status (independently from the sex of the dogs) and their occurrence (% of ’Similar’ responses were lower than 24.7 and % of ’N/A’ responses were lower than 16.1, see Descriptive statistics).

Binomial tests using Age Status on the full sample and comparison with Dominance Status

We tested the age related hypothesis suggested by Bradshaw, Blackwell & Casey (2016), by using the response of the owner to Age (item 19, “Which of your dogs is older?”), to assess differences between dogs allocated an “older” or “younger” status (dogs which were “Similar” in age, or that were marked “N/A”, N = 72, were excluded). Next, we used two-sample tests for equality of proportions with continuity correction in order to determine which factor (Dominance status or Age status) best explained the behavioural and demographic differences between the dogs. This test is used to compare two observed independent proportions. The test statistics analysed by this procedure assume that the difference between the two proportions is zero under the null hypothesis.

Binomial tests on the mixed-sex and same-sex dyads

In order to examine any effect of the dyad composition on dominance status allocation, we created subsets of data including mixed sex dyads (N = 491), and same-sex dyads (N = 512), and ran additional binomial tests to inspect possible associations for items 2 –21. We again adjusted for multiple comparison using Bonferroni correction and lowered the significance level to 0.0025.

Correlation of the items

We used the binary behavioural categories: ”which of your dogs’ expresses the behaviour more”, the dominant = 1 or the subordinate = 0, of the 18 different items: bark, lick mouth, eat first, reward, fight, play ball, greet owner, walk first, resting place, overmark, defend group, smarter, obedient, aggressive, impulsive, size, physical condition, and age, and correlated them using a Pearson Correlation. For this analysis we only used dyads where we had no missing information (N = 215). Results are displayed and discussed in the Supplemental Information.

Differences in the number of dominance related behaviours expressed in dominants and subordinates

We created a “dominance score” by summing all the items that were significantly associated with a “dominant” status for each dog in every dyad. Then we created a “difference score” by subtracting the subordinates’ dominance score from the “dominants” for each dyad. The difference score was then used as the response variable in a general linear model to identify the key variables associated with the difference score. All possible interactions between dominant sex (male or female), subordinate sex (male or female), dominant neuter status (intact or neutered), subordinate neuter status (intact or neutered) and dominant age (older or younger) were entered into the model. We also included the main effect of the order the dogs were entered into the questionnaire (Dog A first or second). We included only the dyads where an asymmetry in dominance was detected by the owner (N = 931). For more details, please see the Supplemental Information.

Results

Descriptive statistics

Eighty-seven percent of owners indicated that their dogs differed in social status, 10% perceived them as similar, and 3% marked the question as “N/A” (Fig. 1). Some items were unreliable for differentiating between the dogs. For example, 30.1–35.1% of the dyads were reported to be similar in greeting the owner, smartness, and physical condition. Other items were relatively difficult for the owners to assess; 16.2–24.3% of owners did not specify a particular dog for the items lick mouth, fight, overmark, and aggressive, Fig. 1).

Binomial tests using Dominance Status on the full sample

We tested which items (from items 2–21) were associated with the perceived dominance rank (item 1). Eleven different dog-dog or dog-owner oriented behaviours, five behavioural/personality traits and five demographic factors were examined. The binomial tests revealed that dogs rated as dominant usually (1) have priority access to certain resources such as food, rewards, resting places; (2) are perceived as undertaking specific tasks, such as “guard” the house through barking more, walk in the front during walks (i.e., “leading” the group), defend the group in case of perceived danger; (3) display dominance: win more fights, mark over the other’s urination, and more frequently accept that the other dogs lick their mouth; (4) have characteristic personality traits: are smarter, more aggressive and impulsive; and (5) are older than subordinates according to the owners. Physical condition, obedience, sequence of greeting the owner and retrieving balls were unrelated to perceived dominance (z ≥ |5.03|, p < 0.0001; see Fig. 2 and Table 2 for an overview of the results).

Figure 2 Percentages with which each characteristic was associated with “dominant” status (numbers in white at the top of each column).

The * next to the item name and blue columns indicate that ”dominant” status was associated with the item after Bonferroni correction (for the Binomial tests all p values are ≤0.0022), and red columns indicate that the characteristic was not associated with the item. Item numbers are in brackets. See Table 2 for more statistical results.

Binomial tests using Age Status on the full sample and comparison with Dominance Status

When we tested the age related hypothesis suggested by Bradshaw, Blackwell & Casey (2016) we found that twelve items were associated with Age status. Six in the same direction as the “dominance” status (bark, lick mouth, fight, resting place, defend group, and smart), and one in the opposite direction, owners found older dogs to be less impulsive, but “dominant” dogs more impulsive. Age but not dominance was associated with five items. Older dogs bark more, play with the ball less, greet the owner less, are in worse physical condition, are larger, and are less often intact than their younger partner dog, according to the owners (p < 0.001). Dominance status was more strongly linked with 11 items in comparison to age status (for statistical details see Table 2).

Table 2 Results of the binomial tests using (A) the owners’ allocation of the dogs to “dominant” or subordinate status (item 1) and (B) “older” or “younger” status (item 19) as the predicted variables and the 21 items. Bold type indicates that status was associated with the characteristic after Bonferroni correction (for the Binomial tests all p values are ≥0.0022).

Two-proportion z-tests were used to determine whether the proportion of “dominant” and “older” dogs were equal for each item. (C) Two-sample tests for equality of proportions with continuity correction in order to determine which factor (Dominance status or Age status) best explained the behavioural and demographic differences between the dogs.

	A. Dominance Status	B. Age Status	Prop. diff	C. 2-sample test for equality of proportions	
Item	Count	Total	Prop	Z	P	Count	Total	Prop	Z	P		X2	P	95% CI	
Bark	547	884	0.619	7.03	<0.0001	512	920	0.557	3.40	<0.0001	0.062	6.953	0.008	0.016	0.109	
Lick mouth	259	737	0.351	−8.03	<0.0001	218	779	0.280	−12.25	<0.0001	0.071	8.669	0.003	0.024	0.120	
Eat first	473	717	0.660	8.51	<0.0001	400	746	0.536	1.94	0.0261	0.124	22.662	<0.0001	0.072	0.175	
Reward	497	684	0.727	11.81	<0.0001	386	714	0.541	2.13	0.0164	0.186	51.141	<0.0001	0.135	0.237	
Fight	606	700	0.866	19.31	<0.0001	443	703	0.630	6.86	<0.0001	0.236	101.920	<0.0001	0.190	0.281	
Play ball	404	793	0.509	0.50	0.7150	349	835	0.418	−4.71	<0.0001	0.091	13.330	<0.0001	0.042	0.141	
Greet owner	352	644	0.547	2.32	0.0100	295	674	0.438	−3.20	<0.0001	0.109	15.194	<0.0001	0.054	0.164	
Walk first	532	795	0.669	9.50	<0.0001	430	824	0.522	1.22	0.1114	0.147	35.819	<0.0001	0.099	0.196	
Resting place	517	716	0.722	11.85	<0.0001	425	754	0.564	3.46	<0.0001	0.158	39.352	<0.0001	0.109	0.208	
Pee	400	669	0.598	5.03	<0.0001	372	697	0.534	1.74	0.0407	0.064	5.465	0.019	0.010	0.118	
Defend group	527	739	0.713	11.55	<0.0001	437	760	0.575	4.10	<0.0001	0.138	30.545	<0.0001	0.089	0.187	
Smart	433	665	0.651	7.76	<0.0001	410	692	0.592	4.83	<0.0001	0.059	4.710	0.030	0.651	0.593	
Obedient	415	838	0.495	−0.24	0.6221	477	879	0.543	2.50	0.0063	−0.048	3.679	0.055	−0.096	0.001	
Aggressive	524	762	0.688	10.32	<0.0001	392	780	0.503	0.11	0.4572	0.185	53.997	<0.0001	0.136	0.235	
Impulsive	512	908	0.564	3.82	<0.0001	313	952	0.329	−10.53	<0.0001	0.235	103.120	<0.0001	0.190	0.280	
Size: heavier	497	929	0.535	2.10	0.0178	575	999	0.567	5.43	<0.0001	−0.032	3.051	0.081	−0.086	0.005	
P Cond: Better	353	687	0.514	0.69	0.2461	209	734	0.285	−11.63	<0.0001	0.229	76.941	<0.0001	0.175	0.280	
Age: Older	615	931	0.661	9.77	<0.0001											
Sex: Male	427	927	0.461	−2.36	0.0090	503	990	0.508	0.48	0.3168	−0.047	4.128	0.042	−0.093	−0.002	
Sex: Female	576	1078	0.534	2.22	0.0131	556	1128	0.493	−0.45	0.6936	0.041	3.621	0.057	−0.001	0.080	
Neutered	580	1073	0.541	2.63	0.0043	613	1133	0.541	2.73	0.0031	0	0.000	1.000	−0.043	0.042	
Intact	423	933	0.453	−2.82	0.0024	446	985	0.453	−2.93	0.0017	0	0.000	1.000	−0.045	0.046	
Notes.

P Cond Physical condition

Prop Proportion

Prop Diff Proportion difference

95% CI 95% Confidence intervals

Table 3 Results of the binomial tests using the owners’ allocation of the dogs to “dominant” or “subordinate” status (Item 1) as the predicted variable and the 20 items in (A) mixed-sex and (B) same-sex dyads.

Bold type indicates that status was associated with the characteristic after Bonferroni correction (for the Binomial tests all p values are ≥0.0022). Two-proportion z-tests were used to determine whether the proportion of “dominant” dogs in mixed-sex and same-sex groups were equal for each item. (C) We compared the “dominants” proportion of each item of each group using a z score calculation with Bonferroni correction for multiple comparisons. Dominant individuals in same-sex dyads mark over subordinate urinations more often than dominants from mixed-sex dyads (same-sex 69% and mixed-sex 51%). Dominant individuals were more often neutered in mixed-sex dyads in comparison to same-sex dyads (mixed-sex 63%, same-sex 53%).

	A.Mixed-sex Dyad	B. Same-sex Dyad	C. Proportion comparison	
Item	Count	Total	Prop	Z	P	Count	Total	Prop	Z	P	Prop. Diff	Z	P	
Bark	268	424	0.63	5.39	<0.0001*	279	460	0.61	4.52	<0.0001	0.03	0.7815	0.4354	
Lick mouth	227	356	0.64	5.14	<0.0001*	251	381	0.66	6.15	<0.0001	−0.02	−0.6011	0.5485	
Eat first	214	348	0.61	4.23	<0.0001*	259	369	0.70	7.70	<0.0001	−0.09	−2.456	0.0139	
Reward	234	329	0.71	7.61	<0.0001*	263	355	0.74	9.02	<0.0001	−0.03	−0.8678	0.3843	
Fight	289	338	0.86	13.00	<0.0001*	317	362	0.88	14.24	<0.0001	−0.02	−0.8011	0.4237	
Play ball	192	383	0.50	0.00	0.5000	212	410	0.52	0.64	0.2605	−0.02	−0.4438	0.6599	
Greet owner	182	314	0.58	2.77	0.0028	170	330	0.52	0.50	0.3102	0.06	1.6426	0.1010	
Walk first	253	375	0.67	6.71	<0.0001*	279	420	0.66	6.68	<0.0001	0.01	0.3105	0.7566	
Resting place	246	340	0.72	8.19	<0.0001*	271	376	0.72	8.51	<0.0001	0.00	0.0831	0.9362	
Overmark	177	346	0.51	0.38	0.6857	223	323	0.69	6.79	<0.0001	−0.18	−4.7143	<0.0001	
Defend group	255	362	0.70	7.73	<0.0001*	272	377	0.72	8.55	<0.0001	−0.02	−0.5127	0.6101	
Smart	205	321	0.64	4.91	<0.0001*	228	344	0.66	5.98	<0.0001	−0.02	−0.6532	0.5157	
Obedient	202	404	0.50	0.00	0.5198	213	434	0.49	−0.34	0.6671	0.01	0.2666	0.7872	
Aggressive	240	359	0.67	6.33	<0.0001*	284	403	0.70	8.17	<0.0001	−0.04	−1.076	0.2801	
Impulsive	252	435	0.58	3.26	0.0005*	260	473	0.55	2.12	0.0172	0.03	0.8994	0.3681	
Size: Heavier	234	450	0.52	0.80	0.2115	263	479	0.55	2.10	0.0177	−0.03	−0.6973	0.4839	
P Cond: Better	168	325	0.52	0.55	0.7471	185	362	0.51	0.37	0.6819	0.01	0.1538	0.8808	
Age: Older	296	455	0.65	6.38	<0.0001*	319	476	0.67	7.38	<0.0001	−0.02	−0.6319	0.5287	
Neutered	310	491	0.63	1.97	0.0488	270	512	0.53	1.86	0.0629	0.10	3.3347	0.0009	
Sex: Female	282	491	0.57	3.25	0.0006*	294	512	0.57	3.31	0.0004	0.00	0.0038	1.0000	
Notes.

P Cond Physical condition

Prop Proportion

Prop Diff Proportion difference

Binomial tests on the mixed-sex dyad sample

In mixed-sex pairs where a dominant was indicated (N = 491), 51% of males and 67% of females were neutered. Females were more often reported as dominant over males (57% females, binomial test z = 3.25, p < 0.001). If we compare dominant females with dominant males in order to help determine what factors might explain why more females are dominant than males: results indicate that 56% of dominant females were older than their partner, in comparison to 66% of dominant males. Thus, older age does not explain the prevalence of dominant females. When a female was rated as the dominant individual, she was more often neutered than when the male was the “dominant” (female neutered 72%, male neutered 51%). “Dominant” males more often (90%) marked over their female partners (while 20% of “dominant” females marked over submissive male partners), defended the group in case of perceived danger, and they were often larger in size than the female “subordinate”. Refer to Tables 3, 4, and the Supplemental Information for more information.

Table 4 Comparison of male and female “dominants” in mixed-sex dyads.

In order to determine whether there were differences between the dominant males and females in each item measured, we compared the dominants proportion of each group (dominant male and dominant female in mixed-sex group) using a z score calculation. Results are displayed for mixed-sex dyads by the sex of the dominant. Bold type indicates that social status was associated with the characteristic after Bonferroni correction (for the Binomial tests all p values are ≥0.0026).

	Dominant female	Dominant male		Proportion comparison	
Item	Count	Total	Prop	Count	Total	Prop	Prop Diff	Z	P	
Bark	162	248	0.65	106	176	0.60	0.05	1.07	0.2846	
Lick mouth	125	201	0.62	102	155	0.66	−0.04	−0.70	0.4839	
Eat first	128	201	0.64	86	147	0.59	0.05	0.98	0.3271	
Reward	137	187	0.73	97	142	0.68	0.05	0.98	0.3271	
Fight	177	205	0.86	112	133	0.84	0.02	0.54	0.5892	
Play ball	111	220	0.50	81	163	0.50	0.01	0.15	0.8808	
Greet owner	96	175	0.55	86	139	0.62	−0.07	−1.25	0.2113	
Walk first	137	222	0.62	116	153	0.76	−0.14	−2.87	0.0041	
Resting place	154	197	0.78	92	143	0.64	0.14	2.82	0.0048	
Overmark	39	193	0.20	138	153	0.90	−0.70	−12.93	<0.0001	
Defend group	127	212	0.60	128	150	0.85	−0.25	−5.22	<0.0001	
Smart	118	183	0.64	87	138	0.63	0.01	0.27	0.7872	
Obedient	114	228	0.50	88	176	0.50	0.00	0	1.0000	
Aggressive	126	206	0.61	114	153	0.75	−0.13	−2.66	0.0078	
Impulsive	158	255	0.62	94	180	0.52	0.10	2.03	0.0424	
Size: heavier	106	257	0.41	128	193	0.66	−0.25	−5.27	<0.0001	
Physical Condition	89	184	0.48	79	141	0.56	−0.08	−1.37	0.1707	
Age: Older	159	258	0.62	137	197	0.70	−0.08	−1.75	0.0801	
Neutered	203	282	0.72	107	210	0.51	0.21	4.78	<0.0001	
Notes.

Prop Proportion

Prop Diff Proportion difference

Binomial tests on the same-sex dyad sample

In same-sex pairs (N = 512, 48.5% neutered) there was no significant difference between the number of neutered and intact “dominant” animals (z = 1.86, p = 0.063). “Dominant” individuals were again more often older than “subordinates” (N = 319 dyads, 67% older, binomial test z = 7.38 p < 0.001).

The items that best described owner reported dominant individuals in the full sample remained significant in the same-sex pairs subsample, apart from the item impulsive, which did not differ between subjects rated as dominant and subordinate. Owners reported that 73% of “dominant” females and 64% of “dominant” males marked over their “submissive” same-sex partners. More results can be found in Table 2.

Difference in the number of dominance related behaviours expressed in dominants and subordinates

The dyad that showed the greatest relationship difference (difference score) between the “dominant” and “subordinate” individual (and therefore the clearest status difference) was in a mixed sex dyad when an intact male was considered as dominant over an intact female. For more results see the Supplemental Information.

Discussion

Our aim was to understand how owners interpret dominance in dog dyads living in their households and to determine whether our psychometric tool measures dominance as defined in ethology. We found that the majority (87%) of owners labelled one of their two dogs as dominant. Perceived dominance status was characterised by fighting ability, submission, competitive ability, subjective resource value, personality, specific roles, and older age. Only thirteen percent of owners were unable to determine a clear rank order between their dogs. This may be because: (1) the dogs have a non-interactive relationship (the partners co-exist without social interactions, i.e., they avoid each other), or an ‘egalitarian’ relationship (the partners affiliate regularly, e.g., play with each other, without agonistic behaviour or exhibiting dominance; Trisko, Smuts & Sandel, 2016); (2) the dogs may not have lived together long enough to form a clear rank order; (3) the owner might actively work against the dogs displaying dominance behaviour (e.g., chasing away the dominant dog from the better resting place, not allowing the dominant to feed first, preventing fights, and favouring the loser dog, etc.); (4) the owner does not accept/understand the concept of dominance; and finally, (5) the survey design encourages that the owner makes a selection.

We found that the results from the questionnaire show external validity. Items associated with the perceived dominance corresponded to behavioural markers of dominance identified by Pongrácz et al. (2008), such as fighting, barking, eating first and receiving mouth licking, and markers related to priority of access to resources such as food, rewards and resting places (Schjelderup-Ebbe, 1922). However, items that examined control over a ball and the owner (greeting) did not differ between “dominant” and “subordinate” dogs, which suggests that the subjective resource value (Van Kerkhove, 2004; Bradshaw, Blackwell & Casey, 2009) has probably a greater effect than the perceived rank. Owners also indicated that dogs higher in status overmark lower ranking dogs, similarly to the findings of Lisberg & Snowdon (2011). In the current study “dominants” marked over same-sexed dogs more often than different-sexed dogs, suggesting higher intra- than inter-sexual competition. Dominance was also associated with items theoretically concerning behaviours viewed by humans as responsibilities, such as defending and leading the group. Similarly, Ákos et al. (2014) found that during off-leash walks, dogs rated as dominant by the owners are more often followed by their group-mates that were rated as submissive. Finally, our results confirmed that some personality traits (aggression, impulsivity, and smartness) are associated with reported dominance, as was suggested by Ákos et al. (2014), who found that aggression towards people and controllability was linked to dominance rank and leadership in pet dogs.

According to the literature, dominance ranks are influenced by several confounding factors, such as age and sex. As predicted, older individuals were more often allocated a higher status by owners in the full sample, and in both mixed and same-sex pairs. However, in contrast to the age related hypothesis, which suggests that age better explains the social structure in dog groups (Bradshaw, Blackwell & Casey, 2016), we found that dominance status, as perceived by the owner, was more strongly associated to 11 of the items than age status. Thus, dog age did not explain the occurrence of dominance related behaviours over and above the owners’ estimate of dominance status. Sex was also linked to dominance; in mixed-sex dog dyads, females were perceived by owners as dominant more often than males, even though in 59% of the dyads they were smaller in size than their male partner. This could be related to the fact that dominant females were more often neutered than dominant males. Previous studies have determined that hormonal activity influences inter-dog aggression (Sherman et al., 1996) and aggression has been found to be more frequent in neutered females compared to intact females and neutered males (Wright & Nesselrote, 1987; Scandurra et al., 2018).

Our study has several limitations: (1) We did not measure the dominant behaviour of the dogs, only the dominance perceived by the owner. Therefore, we have no information about convergent validity, whether the ratings of dominance and behaviour reflect actual rates of behaviour. (2) Only relationships between single dyads were examined. Previous work has determined that individuals can and do establish different types of relationships including “friendships”, when paired with different individuals, and these relationships can also change over time, suggesting high social complexity in dogs (Trisko, Smuts & Sandel, 2016). Future studies should examine how individuals’ relationships differ within multi-dog households. (3) We also did not include items on affiliative behaviour in the questionnaire, so it was not possible to classify the dominance relationships further into formal (affiliation and dominance) and egalitarian (affiliated with no dominance) types (Trisko, Smuts & Sandel, 2016). (4) We were not able to examine breed differences in dominance relationships. Dog breeds and breed groups differ greatly in morphology and typical behaviour (Turcsán, Kubinyi & Miklósi, 2011; McGreevy et al., 2012; Starling et al., 2013), therefore the types of relationships between dogs may also be highly dependent on the breed composition of the group (Van Der Borg et al., 2015). (5) Due to time constraints, we applied single item statements to describe personality traits. (6) We investigated the interpretation of dominance only in Hungary, although there could be significant cultural differences (Wan et al., 2009). (7) We asked owners to compare two dogs to each other, which was difficult for some owners as reflected in a large amount of missing values in the dataset. Future studies should aim to collect data for subordinate and dominant dogs separately using for instance Likert scales, which would allow the use of statistical modelling. (8) Finally, future studies should also investigate inter-observer reliability, thus multiple people should rate the dogs in order to reduce the chance that the answers are reflective of one individual’s views and biases.

Whether dominance as perceived by owners is just a by-product of human observation remains to be answered. However, if we assume that our questionnaire measured actual dominance relationships within the dyads, our results show that the age, sex and neuter status of the dyad influences the relationship between the subjects, which has broader implications for the management of dogs within households. Since humans are ultimately responsible for choosing the social partners (human and conspecific) of their dogs, they have a duty to try to ensure that social relationships are as amicable as possible, in order to keep chronic stress levels, and therefore welfare at an acceptable level. For example, in mixed sex dyads, neutered females were often seen as dominant, and showed behaviours that might increase conflict (reflected in the reduced difference score), regardless of age or body size. Competition in the dyad could be reduced, and any possible increase in dominance motivation in females (caused by neutering) would be avoided, if females could be kept intact and the male neutered if necessary (to prevent breeding). Previous research has indicated that a sex/age graded hierarchy is present in dogs (Cafazzo et al., 2010), and as such, owners could reinforce the position of older individuals in the hierarchy in order to reduce competition in the household. Additionally, to prevent conflicts, owners should try to avoid keeping multiple dogs of the same sex and age. Finally, future studies are necessary to determine how owners perceive their own relationships with their dogs. For example, whether owners have different types of relationships with the dogs within their household, and how this might influence the intraspecific relationships between their dogs; a topic which is currently hotly debated. For instance, one study has determined that dogs form similar relationships with both humans and dogs, and that the quality of the bond varies more with the individual partner than between dog vs. human partners; indicating that relationships cannot be entirely attributed to an individual’s personality (Cimarelli et al., 2019).

Conclusion

Owner estimates of dominance rank corresponded to previously established behavioural markers of dominance displays, which supports that dominance relationships are robust and well-perceivable components of companion dog behaviour and owner-derived reports about dominance ranks have external validity (in pre-schoolers see Hawley, 2002 for similar results). However, the results lack convergent validity, because no simultaneous measure of behaviours were taken, and the data represents only one culture.

We conclude that owners of multiple dogs interpret dominance based on specific behaviours, obtaining resources and certain personality traits. We suggest that future studies that wish to allocate dominance status using owner reports should include the following seven items: which dog starts to bark first, eats first, obtains the reward, walks at the front, acquires the better resting place, defends the group, and is more aggressive. Asking which dog wins fights or which dog licks the mouth of the other might also be useful, as both were highly predictive of owner perceived social status if they did occur, in approximately 70% of cases.

Supplemental Information

Supplemental Information 1 Supplemental Information 1

Analyses regarding the correlation of the items, differences in the number of dominance related behaviors expressed in dominants and subordinates, and difference score models R codes.

Click here for additional data file.

Supplemental Information 2 Questionnaire data, questionnaire labels, pivot tables, and difference score analysis

Click here for additional data file.

We thank Borbála Turcsán for her contribution to the design of the questionnaires, and for her helpful comments on an earlier version of the manuscript. Ivaylo B. Iotchev helped with the literature search and discussions during the manuscript writing. Three anonymous reviewers provided constructive comments.

Additional Information and Declarations

Competing Interests

Author Contributions

Animal Ethics

Data Availability

The authors declare there are no competing interests.

Enikő Kubinyi conceived and designed the experiments, performed the experiments, analyzed the data, contributed reagents/materials/analysis tools, prepared figures and/or tables, authored or reviewed drafts of the paper, approved the final draft.

Lisa J. Wallis analyzed the data, prepared figures and/or tables, authored or reviewed drafts of the paper, approved the final draft.

The following information was supplied relating to ethical approvals (i.e., approving body and any reference numbers):

The procedures applied complied with national and EU legislation and institutional guidelines. Participants were informed about the identity of the researchers, the aim, procedure, and expected time commitment of filling out the survey. Owners filled out the survey anonymously; therefore, we did not collect personal data. Participants could at any point decline to participate (See Supplemental Information S1).

The following information was supplied regarding data availability:

The raw data are available in the Supplemental Files.

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
