# Peer review of "Dominance in dogs as rated by owners corresponds to ethologically valid markers of dominance"

_PeerJ, doi:10.7717/peerj.6838_

## Round 0.1 · original submission · Major Revisions

Thank you for your submission to PeerJ. I have now received reviews of your article from three experts in your field, all of whom have provided thoughtful and detailed reviews (see below). While all three reviewers praised the size of your data set and the merit of your research question, two of the three reviewers had serious concerns with your study, analyses, and interpretation of your findings.

I will not reiterate all the reviewers' comments but I did want to highlight a few key points:

1. The reviewers note that while the use of questionnaire data can certainly be informative, for the specific questions that you are asking, the lack of behavioral data collected in conjunction with the survey data limits your interpretations. At minimum this limitation should be acknowledged.
2. I agree with reviewer 1 that your statistical approach is likely causing you to miss factors that are correlated or interact in some way. While you have corrected for the multiple comparisons that you run, I think a more rigorous approach would be to build a series of models to determine which factors (e.g., age, pair-sex composition etc.) best predict the owners' reports of their dogs' behavior.
3. All three reviewers also provided helpful suggestions as to how you might improve the framing of your paper, through edits to your introduction and discussion.

Some of the concerns raised by the reviewers preclude publication of this version of your article. However, I would like to offer you the opportunity to revise your manuscript and to address these concerns. As you do, please consider each of the reviewers' comments carefully and respond to them fully.

Thank you for considering PeerJ.

Reviewer 1 ·

Basic reporting

The manuscript is overall well-written with regards to the language used. However, while the proposed goal is to address the misuse of the concept of dominance (which in itself is an interesting goal) the authors themselves are inconsistent in the terms they use in their introduction. Specifically, while the authors first appear to distinguish between ‘agonistic’ and formal dominance (line 54), they do not then clearly state which one they refer to. Similarly, at times they appear to use the term ‘dominance’ interchangeably with ‘aggression’ (e.g. line 75-76).

Dominance as an ethological concept refers to an individual’s position within a social group. ‘Dominant’ individuals have higher social positions than ‘subordinate’ individuals. A number of behavioural markers (ritualistic displays, behaviours and outcomes of fights) permit researchers to categorise individuals as ‘dominants’ or ‘subordinates’.
To improve reading, I suggest that the authors re-organize their introduction, to clearly state, form the beginning the various ethological definitions of all relevant concepts (dominance, dominance hierarchy, agonistic behaviours and displays etc.). Then, explain what parameters have been shown to covary with dominance status (age, sex etc) and then explain how these concepts are misused and misunderstood. This way, the ‘gap’ in knowledge will have been logically identified and the aims of the paper can be clearly related to that issue.

The background provided is good. Figures and tables are fine. However, the raw data provided is unclear. Particularly the 'questionaire' sheet, as we have no way of understanding what these values correspond to.
The discussion is fine, although somewhat lengthy with no clear development of the implications of the results. The main question as to whether owners have a good or not understanding of what dominance really is, and whether they can properly assess it in their dogs is not answered. This is partly because the study design and analyses do not allow to do so (see below for more details)

Experimental design

The effort taken in collecting the data is commendable, and so is the dataset gathered. The use of a questionaire, although having limitations, is also fine in the context of this study. However, the overall analysis requires significant improvement in order to allow drawing conclusions that are pertinent for the research question.

The aim was to assess whether owners correctly perceive dominance status in their dogs. In order to do so, the test should be to first objectively define each dog’s dominance status (through ethological/biological measurements) then assess whether the distribution matches owners answers. However, this is not what the authors did. Instead they tested whether the distribution of dogs into dominant vs subordinate categories differed from random. All that this can tell us is that owner-perceived dominant/subordinate dogs perform more or less specific types of behaviours.
Furthermore, as it stands, all items are analysed independently from one another, while in truth you would expect them to be correlated (as at least some of them should be reflecting dominance status). Thus, it would be more appropriate to used GLMMs with binomial distribution (or multinomial logistic regression so that the ‘similar’ category can also be included) to assess how and to what extent the different items predict owner-attributed dominance status.

The supplementary analysis has some potentially valuable results (such as the relationship between behavioural measures of dominance and owners’ reported dominance status), however, the rational for the difference in rank analysis is not provided, and the explanations are not detailed enough to understand what was done and why. Specifically, if a rank difference variable was calculated, then there should be only one value per dyad, so how can “dominants” have higher dominance difference scores?? Furthermore, if the aim was to determine what key variable (i.e. sex, age, neutered status) predicted the size of the score difference, they why create combinations of these parameters? It would be better to include these variables as fixed effects (i.e. age effect as ‘same’ or ‘different’ etc) and include relevant interactions

Validity of the findings

As it stand the validity of the results cannot really be assessed as the statistical analyses need to be improved.
The dataset is commendable, as such with improved analyses the results should be interesting.

Additional comments

Line 30. ‘may be’ instead of ‘maybe’
Line 39. Which ‘psychologists’ and ‘authors’ are you referring to? Please give a reference earlier on.
Line 43. It is unclear what inconsistencies there are. You should contrast how other authors define dominance before making this statement.
Line 54. You state that de Waal distinguishes between agonistic and formal dominance. Is that the stand that you also take? If so, you should keep consistent in the reminder of the ms and specify when you refer to either of these categories. As it stands it is unclear what your stand is.
Lines 63-64. A reference listing like this is not very informative. Can you please, rather explain provide details on what arguments and counter-arguments the different sides propose.
line 75-76. Do you not mean ‘aggression’ rather than ‘dominance’ here? Or are you implying that mixed-sex groups do not have dominance hierarchies?
Line 151-152 appear to be repeats of lines 148-150.
Line 200. The wording is wrong. Binomial tests do not allow you to assess whether a factor predicts another one. It only allows to assess whether the sample distribution statistically differs from a theoretical distribution.

Reviewer 2 ·

Basic reporting

This manuscript is well-written and has an incredibly comprehensive literature review. The raw data, figures, and supplementary analyses are all clear. My only comment is that the 6-item survey the authors suggest for future work could be more easily accessible. It currently is buried in the conclusion section, but it would be nice to have a separate table that lists the 6 questions they suggest (or some way of flagging which questions they suggest in their current Table 1), and perhaps also the additional 2 questions that might prove informative (i.e., which dogs wins fights and which dog licks the owner's mouth). Building on this suggestion, it would be useful for the authors to provide a bit more explanation of why they suggest these specific 6 questions, either in the table legend or in the text of the manuscript itself. I'm assuming it's because they were marked the highest proportion for the dominant dog, but it would be helpful to clarify this so that people can quickly use, understand, and cite the 6-item survey in their future work.

Experimental design

This paper covered original research that is both highly relevant and meaningful, and the method used was appropriately rigorous. However, my one comment is the research question could have been better defined in the introduction. Although the immediate goal of the survey (to determine owner's perceptions of dominance in their dogs) was clear in the method section, the way this linked to a broader goal was not always clear.

In particular, at the beginning of the introduction (lines 29-36) it sounded like the authors were arguing that owners do not have accurate perceptions about dominance in dogs. This left me wondering what a survey with owners could tell us about dominance in dogs since their evaluations on the survey would necessarily be based on misperceptions. Then the following paragraphs talked about other theories of dominance from prior work. All of this is certainly interesting and important literature review, and I'm not suggesting the authors remove this. Instead, I would suggest the authors add one paragraph at the end of the introduction that clearly lays out the goals they have for this survey, both in terms of what we can learn about (1) owner perceptions and (2) actual dominance in dogs (e.g., the age effect the authors discuss, in addition to other research they review in the introduction).

Validity of the findings

An impressive amount of data was analyzed in this study, and the authors did a fantastic job of explaining their analyses and correcting for multiple comparisons throughout. See my note in the "Basic Reporting" section, though, for a way to make the "Difference in ranks" analyses more transparent in the main text. The conclusions are well-stated and link the results of the current study to a wide range of previous research. Any future researcher examining dominance in dogs will find this paper an incredible resource.

My only suggestion here is that the authors more directly mention or address the issue that this is one culture's (i.e., Hungary) impressions of dominance in dogs, so it might be interesting to see how these inferences about dominance differ across other countries.

Reviewer 3 ·

Basic reporting

Paper is mostly well structured and written. The authors write in such a manner that a broad audience can follow and understand what is presented. The cited literature is appropriate as is the presentation of results.

My primary problem with the paper is the discussion (discussed further in my general comments). I feel it could be cut down and more focused on interpreting the results instead of restating them. Use the structure of the conclusions as a guide.

The figures are simple, they could benefit by removal of the background lines. Also, I personally find the tables that are in the supplementary to be more useful than the figures presented. I would consider switching which go in the main text and which go in the supplementary text as, as least for me, it's more important to see the numbers.

Experimental design

A simple study design with a robust sample size. The authors are appropriately cautious with their analyses and though they multiple binomial tests, they use a Bonferroni correction to adjust for p-inflation. As I mentioned in my general comments, it would have been valuable to have the dogs rated by multiple people. A limitation that requires addressing.

There are no ethical concerns and the methods are well described.

Validity of the findings

Data is clearly labelled in Excel sheet and makes sense with how it's formatted. Statistics make sense. As stated above, my primary problem with this paper is the discussion section where not enough interpretation is done, nor is the focus appropriate. More details including line comments can be found in my general comments.

Additional comments

Review of “How do owners perceive dominance in dogs?”

Abstract:
Clear, concise, and easy to follow.

Introduction:
Line 30: Change “maybe” to “may be”. Also, are they unsure or do they simply misunderstand? It’s been my experience that people feel strongly that they know what dominance is, but they are often wrong.
Line 34: Interesting example, though if you’re to speak about abusive training technique I would say “negative reinforcement and positive punishment” and adjust the sentence as Casey et al. looked at both those in their studies and positive punishment has clear abusive connotations.
Line 49: A good point and one that can be leveled against the animal personality literature as a whole as “Dominance” has been used to describe personality domains across primates as well.
Line 89: Adjust to: Pongrácz et al. (2008)
Line 94: Interesting, what does it say about dog dominance that that more dominant individuals copied people more?
Line 99: Perhaps adjust to “In this study we asked owners…”
Overall a well cited and clearly structured introduction.

Methods and materials:
Line 136: Please indicate the version of R used.
I don’t have any other comments. This section is clearly laid out and the statistics make sense and are fairly conservative thanks to the authors’ dedication to Bonferroni corrections.

Results:
Line 178: I don’t find these descriptive statistics overly informative, I wonder if they’d be better as a table in the supplementary materials.
Line 183: The way this is reported is rather awkward. I would like to see it adjusted in a way that mean and SD can be reported as: (mean ± SD = 16.1 ± 8.6) or something equally as readable.
Line 202: Typos: but „dominant”
One problem with the study is that multiple owners weren’t asked to assess the dogs. Given how short the survey is, it seems like this would have been beneficial to reduce the chance that answer are reflective of one owner’s feelings rather than an accurate measurement of observed behavioral patterns.

Discussion:
Line 239: In your introduction you mention that owner assessments of dominance are not novel, perhaps a more tempered first line if the introduction is called for.
Line 240: A hierarchy of two animals isn’t much of a hierarchy and given that you asked owners to only look at two dogs when some had multiple, how accurate is the hierarchy? This makes for a more artificial hierarchy.
Line 243: This is first you specifically say that personality traits were examined but you don’t identify which items or mention this is in the methods. Please bring this up earlier in the paper to make it clear.
Line 244: I would reduce the amount of repetition of percentages and the exact results and focus on interpretation.
Line 263: This is a tough one, if they don’t understand dominance how can you be sure your other rates did as well? Perhaps they just answered based on misinformation and perhaps the accompanying answers are based on this as well, hence the match between dominance status and other items.
Line 273: What is the importance of this difference? Is there an interaction between these differences that explains this? I don’t find this overly compelling because essentially you’ve got a survey where, it’s psychometrically valid, dominance should go with specific behaviors.
Line 286: Again, what does this suggest, what’s the importance?
Line 340: Adjust to: Bonanni et al. (2010)

I’d like to see the discussion cut down and focused on the meaning of these findings. There’s too much time spent reviewing the results again and too little focus on the interpretation and relationship with the rest of the field. I don’t get a feeling for the overall message of the paper and importance. The authors also spend no time considering the validity of their survey. Without behavioral methods I’m not convinced that the survey is actually measuring dog dominance even though the authors spend a lot of time interpreting it in relation to behavioral studies. First, I think the author should discuss how valid their survey is based on their results, then pick the most relevant points from their results to interpret with the broader field, then the limitations of their methods.

Conclusion:
Line 382: I feel like this is the line I was looking for throughout the discussion. I would start the discussion with this focus and argument in support of your survey and work from there.
Line 386: Where was this in the discussion? This conclusion is so well written and focused, I think you should layout your discussion like this.

---

## Round 0.2 · Minor Revisions

The three reviewers who reviewed your initial submission have kindly reviewed your revised submission. All three praised the effort you have clearly shown in improving the strength of your article and enhancing its clarity. That said, all three still have some outstanding concerns. Please respond to each of their comments/suggestions as you prepare your article for resubmission (please note that Reviewer 1 also attached a copy of your article with suggested tracked changes). I believe that, given the improvements you have made to your article, if you are able to respond to these final outstanding concerns it will be my pleasure to accept your article for publication in PeerJ.

In addition to the reviewers’ comments, I also have a few points that I would like you to respond to:

Without building a single model that includes both relative age and relative rank, I am not certain how you can make some of your conclusions (see also Reviewer 3’s comments regarding analyses). For example, on line 263 you state “In mixed-sex pairs (N = 491), females were more often dominant over males (57% females, binomial test z = 3.249, p < 0.001) according to the owners.” Might this be driven by age moreso than sex? Or Perhaps another variable? Importantly, you cannot determine with your current analyses. Without including multiple variables in a single model you cannot determine which is the best explanatory variable (or if multiple are, or if there are interaction effects). At the very least, if you must keep your analyses separated out by variable (age, sex, intact-status, dominance etc.) you should first explore your data set and report if there were or were not any unintentional biases in your data set such that, for example, in mixed-sex pairs, the females were more often also the older dog, or that neutered animals were more likely to be older animals.

In your introduction you use a lot of very short paragraphs (some are single sentences). Doing so breaks up the flow of your writing, making it feel a little choppy. As a consequence, I found it extremely hard to follow your arguments. I suggest you carefully consider your paragraph breaks. Each paragraph should describe a single, clear concept/goal/theory (i.e. each paragraph could be summarized by a single sentence). Consider the points you are making with each and how best to construct your writing as you revise your article.

Also, regarding your introduction, I agree with Reviewer 2 that you could focus your discussion more on owner perceptions than you currently do given the focus of your study. And, as Reviewer 3 notes, I think you could afford to provide a broader context of your findings in your Discussion in addition to providing an overview of what you found specifically.


Editorial suggestions:
I think the second half of your opening paragraph (lines 35-43 starting “Based on observations of macaques…”) does not add anything to your paper and could be deleted.

Line 56: I am not sure what “physically apt” means. Do you mean “physically able”?

Line 182: I suggest you put the question (do dogs that the owners classify as “dominant” show certain behaviours more or less often than their subordinate partner) in quotation marks. This suggestion applies throughout your article where you include questions (e.g., see also line 211)

As noted by Reviewer 3, please ensure that all of your graphs include clear x-axis and y-axis labels. These are unclear or missing from your figures. For example, for Figure 2, the y-axis is simply labeled “%” and the x-axis does not have a label at all. Furthermore, while the y-axis is labeled “%”, the caption describes the data as “proportions” not “percentages” (from looking at the y-axis scale, it seems that the latter is correct). I also agree with the reviewers that Figure 1 is unclear and that the colors are not described in the legend.

The descriptive summary of your data (lines 158-160) is confusing and I think might be better reported in a table. Do you consider neutered dogs as neither male nor female or is it that 23% of pairs were male-male, 28% were female-female and 49% were male-female pairs? Similarly, when describing the individuals, you note 47% were male and 54% were neutered, but what % were female? Is it that 47% were male, 53% were female and, separately, of all the dogs, both male and female, 54% were neutered? (Also, note that in this section, and elsewhere in your article, you switch how you present your percentages, some with decimal points, and some without. Please be consistent.)

For table 1, please indicate which items were used in previous studies and cite the references in the legend. Additionally, I’m not sure that highlighting which items you plan on using in future research is necessary. Lastly, I do not know what you mean by “Items marked by * might also be useful if they occur (in the current study this was the case for approximately 70% of dyads).”

Throughout your manuscript, please indicate where you would like your figures and tables to appear e.g.
“Items 2-4 and 6 were the same as those used in (Pongrácz et al., 2008). In the case of items 20 and 21, the owner could also indicate “both” or “neither” dogs (Table 1).
INSERT TABLE 1 ABOUT HERE
Statistical Analysis
Analyses were performed in SPSS 22.0 and R 3.3.2. Descriptive statistics were calculated for the sample and summarised in the results section.”

Reviewer 1 ·

Basic reporting

The manuscript is much improved regarding the clarity of the the aims and methods and validity of the study. I am happy to say all my previous concerns have been addressed by the significant changes made by the authors.
I have made some minor editorial changes and a few comments on the tracked change docx provided.

I re-state the few comments I still have here:

lines 199-204: Repeat of previous paragraph (Lines 182-188). Please, combine.

lines 210-212: As this analysis aims to test the age-related hypothesis, please detail it in its own paragraph with explicit mention of the hypothesis investigated. Also, some further details on what this test does would be good, as it not a test as well known as typical t-tests and others.

line 269-270: Again, as this tests a separate hypothesis (i.e. age related hypothesis) the results should be better highlighted/detailed.

lines 302-306: Could this not be combined with the descriptive statistics paragraph? Or left for the discussion?

Figure 1: What are the colors of the figure coding for?

Experimental design

No more issues

Validity of the findings

No more issues

Annotated reviews are not available for download in order to protect the identity of reviewers who chose to remain anonymous.

Reviewer 2 ·

Basic reporting

The authors addressed all of my concerns in this section. However, I have a few wording suggestions given the extensive nature of these changes (i.e., many of the sections are essentially fully re-written). In particular I want to flag that there are a number of typos, grammatical mistakes, and some confusing wording that have been introduced in this new version. I have flagged several of these below, but I think a more careful editing pass will need to be done on the paper before it is fully ready for publication.

First, the new Results subsection (Items most convincingly linked to perceived dominance) could be worded more clearly as "Seven items are suggested for future work based on their occurrence (% of 'Similar' responses were lower than 24.7 and % of 'N/A' responses were lower than 16.1, see Descriptive statistics) and their significant link with dominance status, independent from the sex of the dogs: bark, eat first, reward, walk first, resting place, defend group, and aggressive. These items are highlighted in Table 1."

Line 51: The words "rather on" should be removed.

Line 86: The wording of the sentence starting on line 86 is hard to follow

Line 108: Extra period

Line 110: comma can be removed here

Line 386: Typo here - some text seems to be missing (or there is extra text that was not deleted)

Experimental design

The changes here are very helpful for clarifying the goal of the study, especially the two new sentences added to the beginning of the abstract and the new sentence added to the last paragraph of the introduction starting on line 332.

My only further recommendation is that it would be helpful to frame the introduction (from the beginning) more in terms of the owner/public's interpretation of dominance early on. The way the introduction is currently written still sounds like it's being set up to test big picture theories of dominance, rather than the owner's perceptions. This could be as simple as adding a sentence or two somewhere in the first paragraph of the introduction (similar to the new sentences the authors added to the beginning of the abstract in this new version).

Validity of the findings

All my concerns in this section were addressed in the revision.

Reviewer 3 ·

Basic reporting

There are problems with the writing throughout this paper and they become increasingly apparent on a second read through. Perhaps it's a stylistic choice or one relating to a software program but there are single sentence paragraphs in some places and walls of text in others. The introduction and the discussion both needs to be cut to down. Given the study is only survey based, which I believe still makes it worthy of publishing, the authors need not write such a long manuscript and draw such concrete conclusions.

Experimental design

The authors have done a better job at acknowledging the limitations of the study but the focus of the paper drifts away from what the questions are. There are problems with the tables and figures that need to be addressed.

Validity of the findings

As before, I find the conclusion to be well written and direct and feel that the rest of the paper would be greatly improved were this clear, concise, and direct approach taken throughout. The discussion itself needs to be cut down and conclusions drawn need to be made more clear. I found that much of the discussion included reiterating the results or focusing on small parts of them rather than bringing them into the broader literature.

Additional comments

Second review of: How do owners perceive dominance in dogs?


Abstract:
Line 13. This is a well-written and compelling first sentence, I think it’s great.
Line 21: Perhaps instead of “have a certain personality” could say “share certain personality traits…”

Introduction:
Lines 31-43: While interesting, I think this this paragraph is too long given that it doesn’t focus on canids. I’m not convinced that domestic dog dominance resembles the more linear and structured dominance of primates, which makes me inclined to think this paragraph needs shortening and possibly moved to later as the paper isn’t about primates.
Line 44: This is the paragraph that belongs as your opening paragraph.
Lines 71-75: This paragraph needs to be better connected to the rest of the introduction. Currently it reads as an abrupt topic change and a short one at that.
Lines 108-110: I can’t tell if it’s how the paper has been set up but this currently reads as a disconnected paragraph. The statement that dominance is not a personality trait, which I agree with, needs to be supported by a citation.
Line 132: Currently reads as awkward, perhaps adjust to, “In this study, we surveyed people that owned multiple dogs. We investigated the relationship between the dogs’ ranks, behaviour, and demography.”
Lines 133-138: I think this can go in the methods for explaining why you chose the behaviors you did.
Overall, I think the introduction could be cut down and more focused. Also some oddities with how the paragraph are broken up, may be something about the word processing program used but should be fixed.
Line 143: Instead of ‘show’, perhaps use display or perform.

Methods:
Line 197: Why did you not include age and dominance status in the same model? You have a strong sample size and dominance and age don’t have to correlate, why not include them both?
ST 1: What do the highlighted portions of the table indicate? Quite a few of the significant correlations have a fairly low r value. The second column where “Pearson correlation. Sig. (2-tailed)” is repetitive, include that information in the table title or elsewhere so you can delete that column. I’m unsure where the dominance correlations are coming from as they’re not presented in ST1, or perhaps they’re under a column that’s not labelled dominance or are they coming from Table 2 in the main text? This needs to be clarified.
SF1: While simple, I find this figure confusing. Perhaps a simple 2x2 table with the reported percentages would make this easier to understand.
Check the supplementary materials for typos. For example, “dominance” is written “domiance". I appreciate the inclusion of the code, it’s a very nice touch for transparency.

Results:
I found the results to be clear and easy to follow but there are errors and confusing aspects of the tables and figures I’d like to see corrected.
Figure 1: It’s not clear what the different colors indicate.
Figures 1 and 2: Please include titles for all axes.
Perhaps instead of # use boldface, I find the use of # to be distracting and using a symbol that indicates numbers may be confusing to others. Stylistic, I know, but I think it may help for readability.
Across the in-text and supplementary results the columns need to be adjusted so all the title fits, either through the file format or shortening the column names.
Table 3: Mention of bold type but I don’t see any bolding. Were there no significant associations after correction or was this is an error?

Discussion:
The discussion goes into multiple topics but it’s not entirely clear how these minor details add to the overall aim of the study. Again, I think the discussion needs to be shorter and more focused.
Line 299: I would prefer to see a full paragraph instead of a hanging line. Though again, this may be something about how the document has be done.
Line 302: There’s a fifth option: the survey design encourages a selection/answer.
Lines 314-326: I found this paragraph to be difficult to follow as it reads as multiple disconnected sentences. I believe the message is meant to be that there are trends seen across studies but this isn’t clear and could be done much more concisely. Perhaps instead review how your results match those of others and explain how this suggests external validity and be specific. Validity that you’re measuring dominance or that the trends you’re finding suggest your results are valid in a wider context?
Lines 333-371: This wall of text needs to be broken up, as it stands I struggled to follow the writing. This is a common theme with this paper and one that needs to be addressed. Many of the discussion sentences feel disjointed and unfocused.
Line 383: I don’t care for this reference back to the introduction, it asks a lot of the reader. Either include the information here or delete the line.
Line 395: I appreciate the work that has been done to be open about the limitations of the study.

---

## Round 0.3 · accepted · Accept

Thank you very much for your submission to PeerJ. I believe that in this latest revision you have more carefully described your data set and its associated limitations, and provided a more connected interpretation of your results to your aims and methods than in your previous submission. Therefore, it is my pleasure to accept your article for publication in PeerJ.

#